# The Impact of Pdcd4, a Translation Inhibitor, on Drug Resistance

**DOI:** 10.3390/ph17101396

**Published:** 2024-10-19

**Authors:** Qing Wang, Hsin-Sheng Yang

**Affiliations:** 1Department of Toxicology and Cancer Biology, University of Kentucky, Lexington, KY 40536, USA; qing.wang@uky.edu; 2Markey Cancer Center, University of Kentucky, Lexington, KY 40536, USA

**Keywords:** eIF4A, mTORC2–Akt pathway, β-catenin pathway, JNK pathway, paclitaxel, doxorubicin, fluorouracil, platinum-containing drug, IGF1R/IR inhibitor

## Abstract

Programmed cell death 4 (Pdcd4) is a tumor suppressor, which has been demonstrated to efficiently suppress tumorigenesis. Biochemically, Pdcd4 binds with translation initiation factor 4A and represses protein translation. Beyond its role in tumor suppression, growing evidence suggests that Pdcd4 enhances the chemosensitivity of several anticancer drugs. To date, numerous translational targets of Pdcd4 have been identified. These targets govern important signal transduction pathways, and their attenuation may improve chemosensitivity or overcome drug resistance. This review will discuss the signal transduction pathways regulated by Pdcd4 and the potential mechanisms through which Pdcd4 enhances chemosensitivity or counteracts drug resistance.

## 1. Introduction

Although many types of cancer cells are initially susceptible to chemotherapy, over time, they can develop resistance to the treatment, leading to relapse. Resistance to chemotherapeutics can be categorized as either intrinsic or acquired. Intrinsic resistance occurs when resistance-mediating factors are present in cancer cells before chemotherapy, making the treatment ineffective from the beginning. Acquired resistance develops during the treatment of cancers that were initially responsive. This resistance arises due to DNA mutations or activation of alternative cellular signaling pathways during treatment. Protein translation control is a crucial mechanism in regulating cellular transformation and cancer development, which has been previously reviewed [1]. However, its impact on drug resistance has not been thoroughly explored. Programmed cell death 4 (Pdcd4) is a tumor suppressor and a protein translation inhibitor, which has been shown to efficiently inhibit cell proliferation, survival, migration, and invasion in various cancer cells [2,3]. Analysis of the Pdcd4 expression level in a panel of NCI-60 cancer cell lines suggested that Pdcd4 protein level contributes to cellular sensitivity to tamoxifen and geldanamycin [4]. In addition, Pdcd4 protein level is also found to be downregulated in recurrent colorectal cancer patients compared to those with non-recurrent colorectal cancer [5]. These findings indicate that Pdcd4 plays a critical role in drug resistance. In this review, we discuss how Pdcd4 regulates protein translation to influence signaling pathways and examine its role in overcoming drug resistance, along with the potential underlying mechanisms.

## 2. Pdcd4 Functions as a Protein Translation Inhibitor

Pdcd4 was initially recognized as a tumor suppressor because its downregulation in promotion-resistant mouse epidermal JB6 cells led to the acquisition of a promotion-sensitive phenotype [6]. Conversely, ectopic expression of Pdcd4 cDNA in promotion-sensitive JB6 cells results in the development of a promotion-resistant phenotype and suppression of tumor growth [7,8]. Subsequently, Pdcd4 was found to exert multifaceted effects, including inhibiting tumor cell invasion and metastasis [9,10,11,12,13], reducing proliferation [14,15,16], suppressing inflammation [17], and promoting apoptosis [18] (Figure 1).

Beyond suppressing tumorigenesis, Pdcd4 has also been shown to inhibit protein translation. Yeast two-hybrid screens reveal that Pdcd4 binds with translation initiation factor (eIF) 4A [19]. Crystallography and point mutation analyses further confirmed that Pdcd4 interacts with eIF4A through two MA-3 domains composed of eight or nine α-helices [20,21,22] (Figure 2). In addition to the MA-3 domains, Pdcd4 protein possesses several intriguing features. First, the Ser residue at position 67 of Pdcd4 can be phosphorylated by Akt (also called protein kinase B) or ribosomal protein S6 kinase (p70S6K) leading to proteasomal degradation of Pdcd4 [23,24]. Second, the Ser residue at position 457 can also be phosphorylated by Akt, facilitating nuclear localization [24]. Third, the Arg residue at position 110 can be methylated by protein arginine methyltransferase 5 (PRMT5) and this methylation impairs the Pdcd4’s tumor suppressor function [25,26]. Lastly, the N-terminal region of Pdcd4 contains two clusters of positively charged amino acids that have been reported to interact with RNA [27,28] (Figure 2).

Pdcd4 binds to eIF4A and inhibits its helicase activity [19]. eIF4A is an ATP-dependent RNA helicase that plays a key role in protein translation initiation by unwinding the secondary structure of mRNA in the 5’ untranslated region (5′UTR), allowing the translation initiation complex to scan the mRNA [29]. Therefore, inhibition of the eIF4A’s helicase activity by Pdcd4 is expected to suppress protein translation, particularly for mRNAs with a secondary structure in the 5’UTR. This idea was supported by the observations that Pdcd4 represses the translation of luciferase reporter with an artificial stem–loop structure, which has a free energy of −44.8 kcal/mol, greater than the one without the structure [18]. In addition, cells treated with the eIF4A inhibitor, silvestrol, showed decreased luciferase activity in the reporter containing the structured 5’UTR of stress-activated protein kinase-interacting protein 1 (Sin1), while no decrease was observed in the reporter lacking the Sin1 5’UTR [30]. These studies indicate that Pdcd4 preferentially inhibits translation of mRNAs with structured 5’UTR. Besides inhibiting translation through the eIF4A-dependent mechanism, Pdcd4 also inhibits translation via an eIF4A-independent mechanism. It has been reported that Pdcd4 directly binds to the c-myb mRNA by interacting with poly A-binding protein, which may inhibit translation elongation [31]. Liwak et al. found that Pdcd4 directly binds to the internal ribosome entry site (IRES) elements of the mRNA for the X-linked inhibitor of apoptosis (XIAP) and B-cell lymphoma-extra large (Bcl-xL), thereby blocking the formation of a translational initiation complex [32]. A recent study utilizing cryo-electron microscopy suggests that Pdcd4 may impede translation by binding to the 40S ribosome, thereby interfering with the formation of the translation initiation complex [33,34]. Shuvalova et al. [35] also reported that Pdcd4 stimulates protein translation termination by interacting with eukaryotic release factor 3, promoting the release of peptides from the ribosome, which may lead to incomplete protein synthesis. To date, several translational targets of Pdcd4 have been identified including Sin1 [30], p70S6K1 [36], p53 [37], c-myb [38], XIAP [32], and Bcl-xL [32]. These Pdcd4 translational targets play crucial roles in governing multiple signaling transduction pathways that regulate proliferation, survival, apoptosis, and drug resistance.

## 3. Pdcd4 Regulates Signaling Pathway to Control Tumorigenesis and Drug Resistance

### 3.1. mTORC2–Akt Pathway

Akt regulates key cellular processes including survival, proliferation, growth, apoptosis, and metabolism [39]. The activation of Akt is also crucial for the drug resistance of many types of cancer, including lung [40] and breast cancer [41]. Akt is frequently activated via phosphorylation at Thr308 and Ser473. Phosphorylation of Akt at Thr308 is primarily controlled by the phosphoinositide 3-kinase (PI3K)–phosphoinositide-dependent kinase 1 (PDK1) signaling pathway, which drives its basal kinase activity. Additionally, Akt can be phosphorylated at Ser 473 by mammalian target of rapamycin complex 2 (mTORC2). Phosphorylation of Akt at both Thr308 and Ser473 leads to the maximal activation of Akt kinase activity [42]. The mTORC2 complex, composed of the core components mTOR, rapamycin-insensitive companion of mammalian target of rapamycin (Rictor), Sin1, and mammalian lethal with SEC13 protein 8 (mLST8), regulates a range of biological functions, including survival, actin organization, and metabolism, by phosphorylating Akt, protein kinase Cα (PKCα), and serum- and glucocorticoid-regulated kinase 1 (SGK1) [43]. Studies have indicated that Pdcd4 knockdown leads to an increase in Akt phosphorylation at Ser473, whereas the overexpression of Pdcd4 results in a reduction in Akt phosphorylation [44,45]. The Akt phosphorylation at Ser473 and mTORC2 kinase activity are also elevated in Pdcd4 nude mouse embryonic fibroblast (MEF) cells [30]. The increased mTORC2 activity in Pdcd4 knockdown cells is attributed to the elevated Sin1 protein translation, which in turn activates mTORC2 kinase activity and promotes cell invasion [30]. Regulation of Sin1 translation by Pdcd4 is mediated by eIF4A since the Pdcd4 mutant, which is defective in binding with eIF4A, barely inhibits mTORC2 activity and cell invasion [30] (Figure 3). In addition to inhibiting Sin1 translation, Pdcd4 has been reported to bind with Rictor, resulting in suppression of mTORC2 activity [46]. It is possible that Pdcd4, by binding with Rictor, prevents the formation of the mTORC2 complex. However, the precise mechanism underlying this interaction requires further investigation. Knockdown or knockout of Pdcd4 upregulates Snail expression and downregulates E-cadherin expression [13]. Conversely, knockdown of Akt reverses the Pdcd4 knockdown-induced increase in Snail expression and restores E-cadherin level [12,29] (Figure 3). These results suggest there is crosstalk between mTORC2–Akt and E-cadherin–β-catenin signaling pathways.

### 3.2. E-Cadherin–β-Catenin Pathway

The β-catenin signaling pathway, particularly in colorectal cancer, plays a key role in regulating cell proliferation, apoptosis, migration, and chemoresistance. β-catenin functions as a transcription factor by binding to members of the T-cell factor (Tcf) family, thereby stimulating the transcription of its target genes [47]. One of the β-catenin targets involved in chemoresistance is the multidrug resistance protein 1 (MDR1). MDR1 belongs to the ATP-binding cassette transporter family and functions by effluxing chemotherapy agents from cells, thereby reducing drug-induced toxicity in cells and contributing to resistance against chemotherapy agents [48,49]. MDR1 is frequently upregulated in cancer cells, leading to the development of resistance to various anticancer drugs [50]. E-cadherin is one of the major regulators of the β-catenin signaling pathway. E-cadherin is a transmembrane protein whose cytoplasmic domain is associated with β-catenin. Under normal physiological conditions, β-catenin released from E-cadherin is rapidly phosphorylated by glycogen synthase kinase 3 (GSK3) and subsequently degraded via the proteasome pathway. However, when GSK3 activity is inhibited, β-catenin remains unphosphorylated and translocates to the nucleus, where it binds with Tcf transcription factors. The β-catenine/Tcf complex then activates the transcription of target genes involved in β-catenin signaling [51] (Figure 4).

As mentioned above, Pdcd4 knockout or knockdown upregulates Snail and downregulates E-cadherin. E-cadherin expression is regulated by zinc finger transcription repressors of the Snail family, including Snail and Slug [52]. Snail binds to a specific E-box (CAGGTG) in the proximal E-cadherin promoter region, recruiting HDAC to modify the local chromatin structure, thereby repressing E-cadherin transcription [47]. Loss of E-cadherin by Pdcd4 knockdown releases E-cadherin-bound β-catenin from the inner membrane to the cytoplasm. The free β-catenin may then evade proteasomal degradation due to the inactivation of GSK-3 by Akt [15]. Subsequently, the free β-catenin translocates into the nucleus, where it activates β-catenin-dependent transcription to stimulate the expression of oncogenes and chemoresistance genes (Figure 4). The mechanism by which Pdcd4 regulates Snail expression remains unknown and warrants further investigation.

### 3.3. JNK–AP-1 Pathway

The transcription factor activator protein 1 (AP-1) is composed of homodimers from the Jun protein family (c-Jun, JunB, and JunD) or heterodimers formed by Jun and Fos proteins (c-Fos, FosB, Fra-1, and Fra-2). AP-1 recognizes the AP-1 binding site [TGA(G/C)TCA] in the promoter region of the target genes and stimulates their transcription [53]. Upregulation of AP-1 components or activation of AP-1 is essential for promoting cell proliferation, transformation, survival, invasion, and chemoresistance [54,55] (Figure 5). The mRNAs of the AP-1 components or AP-1 activity are frequently upregulated following chemotherapy drug treatment [55]. Knockdown of AP-1 components restores sensitivity to chemotherapy agents [56]. One major mechanism by which AP-1 promotes drug resistance is through the stimulation of MDR1 expression [57]. As discussed in Section 3.2, MDR1 is an efflux protein that reduces intracellular drug toxicity. Additionally, AP-1 may contribute to chemoresistance by promoting the expression of the anti-apoptotic protein XIAP, further enhancing cell survival under chemotherapy [58].

Pdcd4 is an inhibitor of AP-1, functioning at least in part by suppressing the activation of c-Jun and/or c-Fos [8,59]. The potential mechanism for suppression of AP-1 activation by Pdcd4 is through inhibition of the c-Jun terminal kinase (JNK) signaling pathway. Pdcd4 knockdown has been shown to upregulate c-Myc transcription factor, which then binds to the promoter of mitogen-activated protein kinase kinase kinase kinase 1 (MAP4K1) and stimulates its transcription [11,60]. MAP4K1 is an upstream kinase of the JNK signaling pathway. The increase in MAP4K1 expression by Pdcd4 knockdown leads to JNK activation and translocation from the cytosol to the nucleus, where it phosphorylates and activates AP-1 [7,8,59,61]. Moreover, Pdcd4 may also bind directly with c-Jun, preventing its phosphorylation by JNK, which results in the inhibition of AP-1 activation [59]. Taken together, inhibition of the JNK–AP-1 signaling pathway by Pdcd4 plays an important role in overcoming chemoresistance.

## 4. Pdcd4 Counteracts Chemotherapy Resistance

Growing evidence links the loss of Pdcd4 to chemotherapy resistance in various cancer types, though the exact mechanism remains unclear. In this section, we will explore the potential mechanisms by which Pdcd4 overcomes resistance to paclitaxel, doxorubicin, fluorouracil, platinum compounds, and IGF1R-IR inhibitors.

### 4.1. Paclitaxel

Paclitaxel, originally isolated from Pacific yew tree, Taxus brevifolia, has been approved by the FDA for treatment of ovarian, cervical, breast, esophageal, lung, and pancreatic cancers, and Kaposi’s sarcoma. Paclitaxel binds to microtubule polymers, stabilizing them and preventing disassembly. This inhibits proper metaphase spindle segregation, leading to prolonged activation of the mitotic checkpoint and ultimately triggering apoptosis [62]. Although paclitaxel effectively inhibits tumor growth in various cancer types, many patients eventually develop resistance after an initial positive response to the treatment.

Pdcd4 has been shown to enhance chemosensitivity and overcome resistance to paclitaxel. Overexpression of Pdcd4 cDNA has been demonstrated to increase the sensitivity to paclitaxel in colon [63] and prostate cancer cells [64]. Conversely, downregulation of Pdcd4 by *Pdcd4* siRNA promotes the resistance to paclitaxel treatment in breast cancer cells [65,66]. Mechanism-wise, Moustafa-Kamal et al. [67] found that Pdcd4 knockdown significantly prolongs mitotic survival in paclitaxel-treated HeLa cells. In contrast, HeLa cells treated with the S6K inhibitor, PF-4708671, exhibit increased Pdcd4 protein level, consequently improving survival in response to paclitaxel treatment. Since Pdcd4 acts as an inhibitor of eIF4A, using the eIF4A inhibitor hippuristanol to mimic the effect of Pdcd4 showed that hippuristanol induces cell death in a dose-dependent manner when combined with paclitaxel. These findings collectively suggest that Pdcd4 plays a crucial role in enhancing paclitaxel chemosensitivity. The combination of paclitaxel and an eIF4A inhibitor, such as hippuristanol, appears to be a promising strategy for overcoming paclitaxel resistance. As mentioned in Section 2, inhibiting eIF4A is expected to suppress translation of a set of mRNAs involved in critical cellular functions. However, the specific downstream targets of eIF4A involved in paclitaxel resistance remain unclear and require further study.

### 4.2. Doxorubicin

Doxorubicin, also known as Adriamycin, is an FDA-approved drug used alone or in combination with other therapies to treat various blood cancers and solid tumors [68,69]. Doxorubicin functions as a topoisomerase II inhibitor. Topoisomerase II is an essential enzyme for DNA replication and chromosome segregation. Inhibition of topoisomerase II leads to a delay in mitosis, which can ultimately result in cell death if prolonged [70]. Studies have shown that Pdcd4 overexpression enhances the sensitivity to doxorubicin treatment in multiple myeloma cells [71]. Additionally, Gonzalez-Ortiz et al. [72] demonstrated that Pdcd4 expression is downregulated in doxorubicin-resistant MDA-MB-231 breast cancer cells, which subsequently reduces the interaction of Pdcd4 and eIF4A, implying that eIF4A downstream targets are involved in doxorubicin resistance. Knockdown of eIF4A in these resistant cells decreases focal adhesion kinase (FAK) expression, suggesting that FAK is one of the eIF4A downstream targets [72]. FAK, a protein tyrosine kinase, regulates cellular migration, proliferation, and survival [73], playing a critical role in doxorubicin resistance. Inhibition of FAK activity has been shown to overcome resistance in doxorubicin-resistant lung and breast cancer cells [74,75]. Thus, Pdcd4 counteracts the resistance to doxorubicin by binding to eIF4A to inhibit FAK activity.

### 4.3. Fluorouracil

Fluorouracil (5-FU) is a uracil analog, where the hydrogen atom at the C-5 position is replaced by a fluorine atom. It has been widely used to treat gastrointestinal, breast, gynecological, and head and neck cancers [76]. The 5-FU analog inhibits thymidylate synthase activity and thereby suppresses deoxythymidine mono-phosphate production required for DNA replication and repair [77]. Additionally, 5-FU can be misincorporated into RNA and DNA in place of uracil or thymine, disrupting normal nucleic acid function [77]. As a result, 5-FU treatment impairs cell cycle progression and ultimately induces cell death. However, resistance to 5-FU frequently develops, reducing the efficacy of the therapy and leading to poor prognosis.

Several reports have shown that Pdcd4 expression is significantly downregulated in 5-FU-resistant colorectal or pancreatic cancer cells [5,78,79], highlighting its importance in 5-FU resistance. The loss of Pdcd4 expression in these resistant cells is mediated by microRNA or long non-coding RNA (LncRNA) inhibition. Conversely, ectopic expression of Pdcd4 in 5-FU-resistant HCT116 and SW480 cells restores their sensitivity to 5-FU treatment. Overexpression of Pdcd4 in 5-FU-resistant colorectal cells induces apoptosis and inhibits invasion by upregulating the pro-apoptotic Bax and downregulating invasion-promoting MMP-2 and MMP-9 [5].

One potential mechanism by which Pdcd4 regulates expression of Bax, MMP-2, and MMP-9 is through the inhibition of NF-κB to suppress their transcription [80]. NF-κB consists of subunits such as p105/p50, p100/p52, p65, c-Rel, and RelB, which form homo- or heterodimers [81]. In the cytoplasm, IκB binds with NF-κB, preventing its translocation to the nucleus. Upon phosphorylation by IKKα/IKKβ, IκB is degraded, allowing NF-κB to translocate into the nucleus and activate the transcription of target genes [82]. Pdcd4 is known to inhibit Akt activation, and Akt knockdown suppresses IKKα/IKKβ activation [15]. Therefore, Pdcd4 likely suppresses MMP-2 and MMP-9 expression through the Akt– IKKα/IKKβ–NF-κB axis.

In addition to the aforementioned mechanism, Hu et al. [79] reported that the overexpression of Pdcd4 inhibits the upregulation of ATP-binding cassette subfamily G2 (ABCG2) in 5-FU-resistant HCT116 cells. This provides another potential mechanism by which Pdcd4 overcomes 5-FU resistance.

### 4.4. Platinum-Containing Drug

Platinum-containing drugs, such as cisplatin, carboplatin, oxaliplatin, ormaplatin, and enloplatin, are commonly used to treat various cancers, including lung, ovarian, head and neck, breast, and brain cancers. However, their effectiveness is often limited by the development of resistance in patients. These drugs exert their anticancer effects by disrupting the DNA repair mechanisms and inducing DNA damage through crosslinking with purine bases, primarily guanine and adenine [83], which ultimately leads to cancer cell death. Pdcd4 has been reported to enhance the chemosensitivity of cisplatin. In a comparison of Pdcd4 expression level across various ovarian cancer cell lines, Zhang et al. found that cells with higher Pdcd4 levels are more sensitive to cisplatin [84]. Overexpression of Pdcd4 further increases cisplatin chemosensitivity, promoting cell death via elevated levels of cleaved caspase-3 and caspase-8 in both cultured cells and mice [84]. On the contrary, downregulation of Pdcd4 reduces cisplatin sensitivity [84,85,86].

Liu et al. reported that loss of Pdcd4 expression leads to the upregulation of MDR1 in cisplatin-resistant HeLa cells [87]. They also found that Pdcd4 associates with Y-box binding protein 1 (YB-1), a transcription factor, and subsequently binds to the MDR1 promoter region to repress its transcription [87]. In addition to transcriptional regulation, Pdcd4 may also regulate MDR1 translation through Akt inhibition. As mentioned in Section 3.1, Pdcd4 inhibits Akt activation by suppressing the translation of Sin1 [30]. Non-phosphorylated YB1 is able to bind with the MDR1 mRNA at the capped 5’-end, which blocks the translation initiation factors’ assembly and thereby suppresses translation of MDR1. Upon phosphorylation by Akt, YB1 reduces its affinity for the MDR1 mRNA at the capped 5’-end, relieving translational repression and allowing MDR1 expression to increase [88]. Moreover, Pdcd4 may suppress MDR1 expression by inhibiting activation of AP-1 or β-catenin, as discussed in Section 3.3 and 3.2, respectively. Further investigation is required to determine whether activation of AP-1 or β-catenin due to Pdcd4 loss contributes to cisplatin resistance.

### 4.5. IGF1R/IR Inhibitors

Insulin-like growth factor 1 receptor (IGF-1R) and insulin receptor (IR) are receptor tyrosine kinases that, upon binding with IGF1, IGF2, or insulin, activate downstream signaling pathways, such as the PI3K–Akt and Ras–ERK pathways. These pathways primarily regulate cancer cell proliferation, survival, and apoptosis [89]. Therefore, inhibiting IGF1R/IR activation is expected to efficiently suppress tumor growth. Two types of IGF1R/IR inhibitors have been developed: monoclonal antibodies and small-molecule inhibitors [90]. Monoclonal antibodies block the binding of ligands to IGF1R/IR, while small-molecule inhibitors compete for the ATP binding site of IGF1R/IR. Although in vitro and animal studies have shown promising results in suppressing tumorigenesis with IGF1R/IR inhibitors, clinical outcomes have shown a lack of survival benefit for patients treated with these inhibitors, along with an increase in adverse events [91]. The inefficacy is likely attributed to the development of resistance.

In an analysis of 27 colorectal cancer cell lines, Pitt et al. classified 16 of them as resistant and 6 as sensitive to the IGF1R/IR inhibitor OSI-906 [92]. Interestingly, the resistant cells exhibit relatively low levels of Pdcd4 protein, while sensitive cells show high Pdcd4 levels, suggesting a correlation between Pdcd4 expression and chemosensitivity to OSI-906 in colorectal cancer cells [93]. Furthermore, overexpression of Pdcd4 in IGF1R/IR inhibitor-resistant cells restores sensitivity to the inhibitor, whereas knockdown of Pdcd4 induces a resistant phenotype [93].

When resistant colorectal or breast cancer cells are treated with IGF1R/IR inhibitors, such as OSI-906, BMS-754807, or GSK1838705A, phosphorylation of p70S6K is induced [36,93,94]. However, Pdcd4 overexpression inhibits this IGF1R/IR inhibition-induced p70S6K phosphorylation. Activation of p70S6K1 has been shown to promote cell survival by phosphorylating mouse double minute 2 (MDM2) [95], an E3 ubiquitin ligase. Phosphorylation of MDM2 by p70S6K1 leads to polyubiquitination and degradation of p53, thereby suppressing p53-dependent apoptosis [96]. Further investigation revealed that Pdcd4 specifically inhibits the translation of p70S6K1, but not p70S6K2, to control the level of p70S6K1 phosphorylation [36]. In mouse models, combination treatment of OSI-906 (an IGF1R/IR inhibitor) and PF-4708671 (a p70S6K inhibitor) significantly reduces IGF1R/IR inhibitor-resistant colorectal tumor growth [93]. Additional studies are needed to assess the efficacy of this combination for clinical application.

## 5. Conclusions and Perspectives

Pdcd4 is increasingly recognized as a critical player in combating drug resistance, primarily due to its role as an inhibitor of protein translation. Loss or downregulation of Pdcd4 is linked with increased drug resistance, while its upregulation enhances cancer cells’ sensitivity to chemotherapeutic drugs. Therefore, Pdcd4 has potential both as a biomarker for predicting treatment responses and as a therapeutic target for re-sensitizing drug-resistant cancer cells. Further investigation into the molecular mechanisms by which Pdcd4 re-sensitizes resistant cancer cells to drugs could uncover new therapeutic targets, offering the dual benefits of suppressing tumor progression and overcoming drug resistance. Additionally, the identification of new agents that can restore Pdcd4 expression or function could eventually lead to clinical applications that boost the efficacy of existing therapies in overcoming resistance.

## Figures and Tables

**Figure 1 pharmaceuticals-17-01396-f001:**
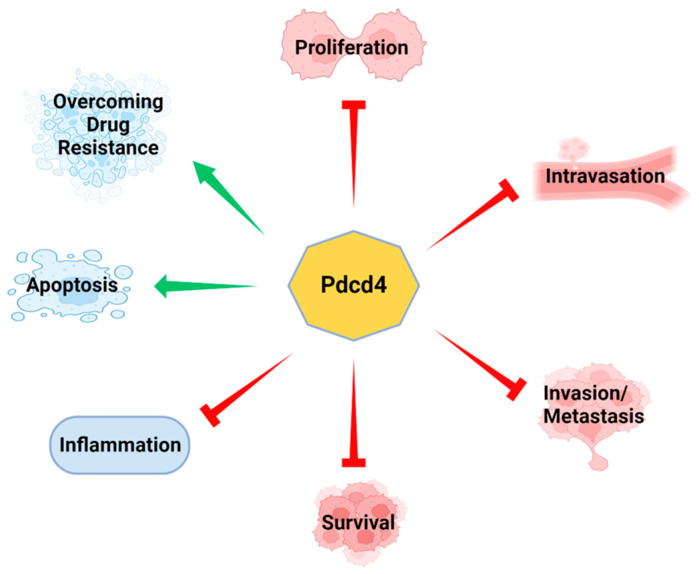
Pdcd4 is a tumor suppressor. Experimental evidence has shown that Pdcd4 suppresses various cancer cell characteristics including inflammation, proliferation, survival, invasion, and metastasis. Additionally, Pdcd4 promotes apoptosis and helps to overcome drug resistance.

**Figure 2 pharmaceuticals-17-01396-f002:**
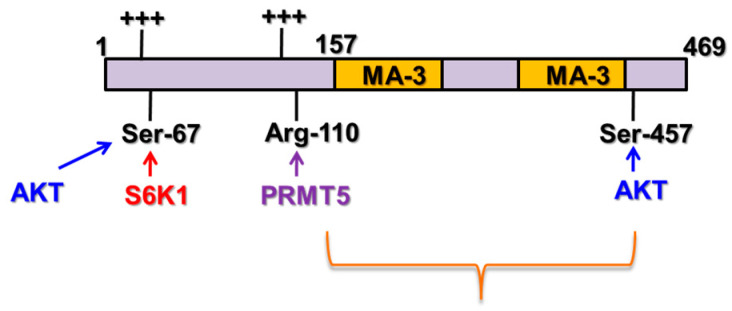
Schematic diagram of the functional motifs of Pdcd4. Two MA-3 domains binding with eIF4A to inhibit protein translation; Ser67 phosphorylated by Akt or p70S6K for proteasome degradation; Ser457 phosphorylated by Akt for nuclear localization; Arg110 methylated by PRMT5 to attenuate the tumor suppression function; the positive amino acid cluster (+++) binding with RNAs.

**Figure 3 pharmaceuticals-17-01396-f003:**
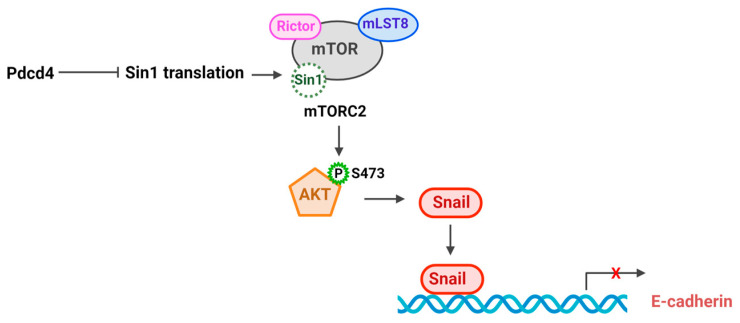
Pdcd4 suppresses mTORC2–Akt pathway.

**Figure 4 pharmaceuticals-17-01396-f004:**
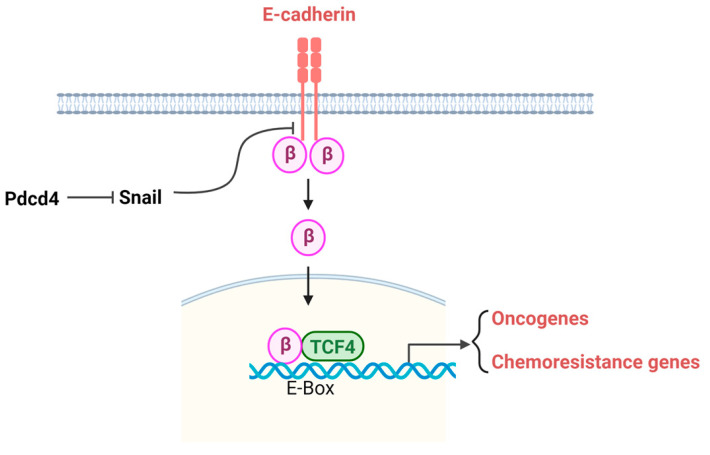
Pdcd4 suppresses the E-cadherin–β-catenin pathway. The transcription repressor Snail suppresses E-cadherin expression, leading to β-catenin nuclear translocation and binding with Tcf4 to stimulate expression of oncogenes and chemoresistance genes.

**Figure 5 pharmaceuticals-17-01396-f005:**
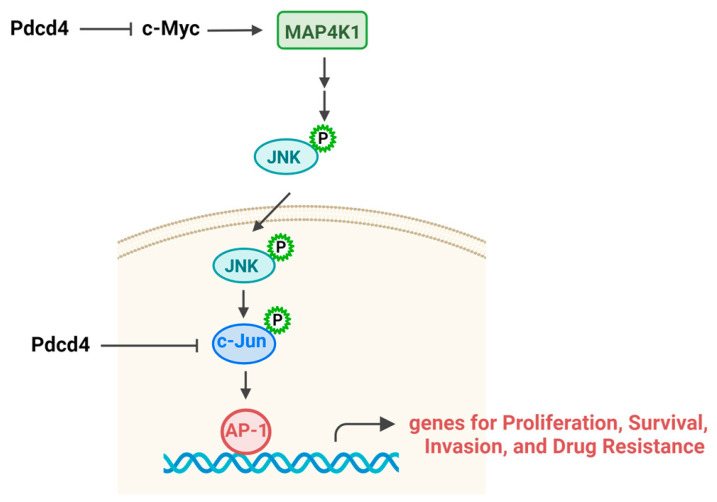
Pdcd4 suppresses JNK–AP-1 pathway. Pdcd4 inhibits MAP4K1 expression or directly binds to c-Jun, resulting in suppression of JNK–AP-1 pathway.

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
