# Peer review of "The Impact of Pdcd4, a Translation Inhibitor, on Drug Resistance"

_pharmaceuticals, 2024, doi:10.3390/ph17101396_

Round 1
Reviewer 1 Report
Comments and Suggestions for Authors
Your paper provides valuable insights into the role of Pdcd4 in cancer drug resistance, particularly in relation to its interactions with various signalling pathways. However, there are several areas where improvements can be made:
- Organization and Clarity: While the sections follow a logical order, some transitions between topics are abrupt. Consider providing clearer links between the distinct functions of Pdcd4 in various signalling pathways and how these collectively contribute to overcoming drug resistance. Additionally, the introduction could better outline the key contributions of the paper to help guide the reader.
- Depth of Discussion: Some pathways, like the JNK-AP-1, could benefit from further elaboration, especially in terms of potential therapeutic interventions that could target these pathways. Highlighting more recent advancements in Pdcd4 research would add further value.
- Language and Readability: The language, while mostly clear, contains complex sentences and awkward phrasing that make certain sections difficult to understand. I recommend moderate editing for grammar and readability. Simplifying some sentences and using more direct language could enhance the flow of the paper.
- References: While you have included a solid foundation of references, consider adding more recent studies on Pdcd4, especially those that explore its role in novel cancer therapies or resistance mechanisms. This would strengthen the paper’s relevance and contribution to the field.
Comments on the Quality of English Language
Some sections are challenging to follow due to awkward phrasing and occasional grammatical errors. Moderate revisions would improve clarity and readability, particularly in complex sentences.
Author Response
- Organization and Clarity: While the sections follow a logical order, some transitions between topics are abrupt. Consider providing clearer links between the distinct functions of Pdcd4 in various signaling pathways and how these collectively contribute to overcoming drug resistance. Additionally, the introduction could better outline the key contributions of the paper to help guide the reader.
Response: Thank you for the suggestion. The manuscript has been revised to improve its organization and clarity.
- Depth of Discussion: Some pathways, like the JNK-AP-1, could benefit from further elaboration, especially in terms of potential therapeutic interventions that could target these pathways. Highlighting more recent advancements in Pdcd4 research would add further value.
Response: The manuscript has been revised to provide a more in-depth discussion of the role of each signaling pathway in drug resistance.
- Language and Readability: The language, while mostly clear, contains complex sentences and awkward phrasing that make certain sections difficult to understand. I recommend moderate editing for grammar and readability. Simplifying some sentences and using more direct language could enhance the flow of the paper.
Response: This manuscript has been edited for grammar and readability.
- References: While you have included a solid foundation of references, consider adding more recent studies on Pdcd4, especially those that explore its role in novel cancer therapies or resistance mechanisms. This would strengthen the paper’s relevance and contribution to the field.
Response: We have included 12 additional references, of which 6 were published within the last 3 years. Please note that there are limited publications on the mechanism by which Pdcd4 enhances chemosensitivity and/or overcomes chemoresistance. We have cited as many recent publications as possible.
Comments on the Quality of English Language
Some sections are challenging to follow due to awkward phrasing and occasional grammatical errors. Moderate revisions would improve clarity and readability, particularly in complex sentences.
Response: Edited to improve the clarity and readability.
Reviewer 2 Report
Comments and Suggestions for Authors
In their review, the authors discussed the correlation between Programmed cell death 4 (Pdcd4) and resistance to various anticancer drugs (doxorubicin, fluorouracil, platinum compounds, and IGF1R-IR inhibitors). In my opinion, this review did not present any novel findings that would make it more useful than other reviews. However, some additional comments were included as follows:
1. The introduction section should provide the rationale for this review, highlighting the scientific gaps in existing reviews and emphasizing the novelty of the current review. This is crucial for capturing the readers’ interest.
2. All abbreviations, including the cell line abbreviation, must be explained before use.
3. Line 88, PRMT% or PRMT5?
4. The figures and illustrations were designed as a copy from other reviews. Where is the novelty of the study?
Author Response
In their review, the authors discussed the correlation between Programmed cell death 4 (Pdcd4) and resistance to various anticancer drugs (doxorubicin, fluorouracil, platinum compounds, and IGF1R-IR inhibitors). In my opinion, this review did not present any novel findings that would make it more useful than other reviews. However, some additional comments were included as follows:
- The introduction section should provide the rationale for this review, highlighting the scientific gaps in existing reviews and emphasizing the novelty of the current review. This is crucial for capturing the readers’ interest.
Response: Thank you for the suggestion. The role of Pdcd4 in tumorigenesis has been reviewed by our laboratory and others. However, this manuscript is the first to systematically review Pdcd4’s role in drug resistance, particularly through the translational control axis. We have added this point in the introduction section.
- All abbreviations, including the cell line abbreviation, must be explained before use.
Response: The abbreviations have checked and corrected.
- Line 88, PRMT% or PRMT5?
Response: This typo has been corrected. Thank you.
- The figures and illustrations were designed as a copy from other reviews. Where is the novelty of the study?
Response: All figures in this manuscript are original and created using BioRender software. As it is a review manuscript, each figure summarizes published data, which may result in some parts of figures appearing similar to those in previously published reviews.
Round 2
Reviewer 1 Report
Comments and Suggestions for Authors
I think this time its correct.
Reviewer 2 Report
Comments and Suggestions for Authors
The authors have improved the review and addressed all the comments. The only comment is in Figure 1, where the title is "Pdcd4 is a tumor suppressor" and the associated explanation must be transferred to the next paragraph.